# The New Insight into the Effects of Different Fixing Technology on Flavor and Bioactivities of Orange Dark Tea

**DOI:** 10.3390/molecules28031079

**Published:** 2023-01-20

**Authors:** Yuanfang Jiao, Yulin Song, Zhi Yan, Zhuanrong Wu, Zhi Yu, De Zhang, Dejiang Ni, Yuqiong Chen

**Affiliations:** 1National Key Laboratory for Germplasm Innovation and Utilization for Fruit and Vegetable Horticultural Crops, College of Horticulture & Forestry Sciences, Huazhong Agricultural University, Wuhan 430070, China; 2Zigui County Agricultural and Rural Bureau, Yichang 443600, China; 3Key Laboratory of Urban Agriculture in Central China, Ministry of Agriculture, Wuhan 430070, China

**Keywords:** Qingzhuan tea, orange dark tea, fixing method, processing, quality, activity

## Abstract

Peach leaf orange dark tea (ODT) is a fruity tea made by removing the pulp from peach leaf orange and placing dry Qingzhuan tea into the husk, followed by fixing them together and drying. Since the quality of traditional outdoor sunlight fixing (SL) is affected by weather instability, this study explored the feasibility of two new fixing methods, including hot air fixing (HA) and steam fixing (ST). Results showed that fixing method had a great impact on ODT shape, aroma, and taste. Compared with SL and ST, HA endowed ODT with higher fruit aroma, mellow taste, better coordination, and higher sensory evaluation score. Physical–chemical composition analysis showed that SL-fixed orange peel was higher than HA- or ST-fixed peel in the content of polyphenols, flavonoids, soluble protein, hesperidin and limonin, while HA has a higher content of volatile substances and contains more alcohols, aldehydes and ketones, and acid and esters than ST and SL. Activity analysis showed that HA was superior to ST or SL in comprehensive antioxidant activity and inhibitory activity against α-glucosidase. Comprehensive results demonstrated that HA has better performance in improving ODT quality and can replace the traditional SL method in production.

## 1. Introduction

Dark tea, one of the main teas in China, has long been a necessity of life for ethnic minorities in the frontier districts. In recent years, it has also been very popular in Hong Kong, Macao, Taiwan, Southeast Asia, and other regions or countries [1]. Studies have shown that dark tea has the functions of weight loss, lipid reduction, blood sugar reduction, anti-aging, antioxidation, and improvement of intestinal function [2]. Orange peel, a by-product of citrus and accounting for 25~40% of total fruit weight, contains polyphenols, flavonoids, synephrine, polysaccharides, essential oils, pectin, limonoids, carotenoids, and other natural ingredients with good functional activity [3]. Among them, flavonoids have strong antioxidant and anti-aging effects [4]; synephrine and other alkaloids have certain effects on cardiotonic, asthma, and shock treatment [5]; limonin and its analogs have anti-cancer, anti-inflammatory, and analgesic effects [6].

In China, citrus tea is a special blended tea made by hollowing out the pulp of fresh citrus, followed by filling tea leaves into the husk, then fixing them together and drying, endowing the citrus tea with both the aroma of tea and the fruity aroma of orange peel and gaining high popularity among consumers [7]. Recent studies have found that citrus tea has significantly stronger in vitro antioxidant and antitumor activities than tea or citrus peel alone, due to the synergistic action of active substances in tea and citrus peel to enhance the functional activity [8]. Studies have also shown that citrus tea has better effects than tea alone in anti-depression and regulating gut microbes [9]. These reports indicate that the combination of citrus and tea may have more health benefits.

Based on modern technology, the process of citrus tea can be divided into two stages: fixing and drying, with fixing as the most important process, inactivating the enzyme through high temperature and reducing further enzymatic reaction [10]. In the traditional citrus-tea production process, fixing is based on outdoor sunlight, but the quality is not stable due to weather changes. In recent years, new fixing methods such as steam and hot air fixing have been developed, but few studies have been performed on the use of these methods in citrus-tea production. In the research of fixing methods for botanical materials, fixing methods were shown to vary in their quality characteristics and antioxidant activities. For example, after different fixing treatments, citrus flowers were found to vary significantly in the contents of polyphenols and soluble sugars, as well as aroma characteristics.

Peach leaf orange is a characteristic citrus germplasm resource in Hubei Province, with sweet flesh and strong rose-like aroma. Its peel contains active ingredients such as polyphenols and flavonoids, which have the same antioxidant and anti-tumor activity as other oranges, and especially help with digestive function. Local people have long had the custom of eating orange peels, such as chopping and pickling them before eating or using them as a cooking condiment. Thus far, few studies have been performed on its application in citrus-tea production. Our preliminary research shows that Qingzhuan tea (a representative dark tea in China) has better functions of weight loss and fat reduction and regulation of the gastrointestinal tract [11]. Therefore, the combination of the two will help to complement and improve the functions, and the development of peach leaf orange dark tea will fill the gap in this field. In recent years, we have paired orange peel with Qingzhuan tea and processed it into peach leaf orange dark tea (ODT), which has a rich fruit flavor and mellow taste and is loved by consumers and very popular among consumers. In this study, the same tea leaves and orange raw materials were used to process ODT through the steps of washing, hollowing out pulp, filling dry tea leaves into husk, spreading, fixing, and drying. In the fixing process, traditional sunlight fixing (SL) was compared with the modern hot air fixing (HA) and steam fixing (ST) methods to explore their effects on ODT quality. On this basis, fixing temperature was further optimized, and the resulting parameters are expected to guide the development of the citrus-tea industry. 

## 2. Results

### 2.1. Effects of Different Fixing Methods on Sensory Quality

The ODT evaluation system was based on the tea sensory evaluation method, but with added coordination evaluation of aroma and taste to reflect the harmony between peel and tea [12]. In taste evaluation, coordination refers to the mutual balance of sour, sweet, bitter, and astringent tastes in each tea soup, i.e., the fusion of orange peel and tea taste. The sensory evaluation results of ODT with different fixing methods are shown in Table 1. All the three ODTs had a relatively high sensory evaluation score (77, 75, and 74.6 marks, respectively), with bright red tea soup, strong fruity aroma, and satisfactory coordination. The three ODTs showed no significant difference in scores of aroma and soup color but notable (*p* < 0.05) difference in appearance and taste.

In terms of appearance, the sunlight (SL) fixed ODT has a yellow-green but uniform peel, while the hot air (HA) and steam (ST) fixed ODTs have greenish and slightly darker peels, indicating a great influence of different fixing methods on orange peel color. Hu et al. (2006) also found that beans treated with hot air had a darker color [13], which was similar to the results of the present study. Overall, the appearance score is significantly (*p* < 0.05) higher in SL ODT than in HA or ST ODT. 

In terms of taste, HA ODT was higher than ST or SL ODT in the scores of sourness, bitterness, astringency, and harmony, achieving the highest overall taste score (37.5, 36.2, and 35 for ST and SL ODT, respectively). Figure 1 shows the radar map of taste factor scores for different fixing methods. 

In terms of sourness, HA showed significantly better performance than ST or SL, and ST was superior to SL. In terms of astringency, SL- and ST-fixed ODTs showed a more astringent taste. In terms of bitterness, HA showed better performance than ST or SL. In terms of coordination, HA showed the highest score, followed by ST, and SL had the lowest score. The coordination performance is jointly determined by various taste factors of the tea soup, so obvious sourness, bitterness, and astringency suggest poor taste coordination. SL and ST were lower than HA in sourness and astringency scores, so their taste coordination scores were also lower.

### 2.2. Effects of Different Fixing Methods on Chemical Composition

The results of the moisture content of tea leaves and peels after fixing in three methods are shown in Appendix A. The main chemical components in the peels treated with different fixing methods are shown in Figure 2A,B. The three fixing methods showed significant (*p* < 0.05) differences in the content of each substance. Specifically, SL-fixed peel was significantly higher than HA- or ST-fixed peel in the contents of soluble protein, polyphenols, flavonoids, hesperidin, synephrine, and limonin. 

The main chemical components of the tea treated with different fixing methods are shown in Figure 2C,D. HA was significantly higher than ST or SL in tea polyphenol content. Compared with ST- or SL-fixed tea, HA-fixed tea had less oxidation of polyphenols in the fixing process and was also lower in the content of the two tea polyphenol oxidation products of thearubin and theabrownin. Meanwhile, ST had the highest content of soluble sugar, probably due to its high temperature, resulting in hydrolysis of polysaccharides in tea to soluble sugar, thus increasing its content. Compared with the peel, the tea was less affected by the different fixing methods.

### 2.3. Effects of Different Fixing Methods on Volatile Component

Based on above sensory evaluation, the three ODTs fixed by the three different methods all showed strong fruity aroma, but they varied in their aroma types, so their volatile components were further analyzed by headspace solid-phase microextraction–gas chromatography–mass spectrometry (HS-SPME-GC-MS). Comparison of NIST database and retention index (RI) values identified 108, 106, and 106 compounds in HA, ST, and SL ODTs, respectively. As shown in Figure 3A,B and Table 2 and Table 3, these species can be grouped into alcohols, aldehydes and ketones, alkenes, acids and esters, and others, with alcohols, alkenes and aldehydes, and ketones accounting for more than 90% of total aroma, hence the main part of ODT aroma.

Alcohols mainly provide pleasant aromas, mostly floral and fruity [14], covering the largest proportion of total HA aroma (39%), followed by SL (38.2%), and ST (31.6%). Among the alcohols, linalool has the highest content, followed by Terpinen-4-ol, carveol, and its isomers. Linalool has an obvious floral aroma and is widely present in various types of tea and also has a high content in citrus fruits [15]. For the three differently fixed ODTs, HA and SL were slightly higher than ST in linalool content. Additionally, HA was also significantly (*p* < 0.05) higher than SL or ST in the contents of terpinen-4-ol, trans-carveol, cis-isocarveol, and cis-carveol. In terms of unique compounds, dihydrocarveol was absent in HA and SL but present in ST.

Aldehydes and ketones, important aroma components in tea, are widely present in tea and fruits and are also important volatile components in citrus tea [7]. For the three differently fixed ODTs, the proportion of aldehydes and ketones in total aroma was similar in the three samples, 28.4% (HA), 27.9% (ST), and 26.5% (SL) respectively. However, HA was significantly higher (*p* < 0.05) than ST and SL in the contents of decanal, octanal, nonanal, (+)-carvone, (Z)-citral, citral, dodecanal, and other substances. 

Alkenes are an important part of tea aroma and widely present in citrus tea [7]. In this experiment, 30, 32, and 30 alkenes were identified in HA, ST, and SL, respectively, and alkenes were more abundant than the other substances. Among the alkenes detected, limonene is a representative aroma compound in citrus fruits and has the highest content, accounting for more than half of total alkenes. In the three ODT samples, ST was significantly (*p* < 0.05) higher than HA or SL in limonene content. In addition, despite a relatively high content of β-cadinene, α-Copaene, and (-)-α-cubebene in alkenes, the three samples showed no significant difference in the content of these substances.

The acid and esters in citrus tea mainly include geranyl acetate, methyl N-methyl anthranilate, and dihydroactinidiolide, which showed a higher content in ST than in HA or SL. However, no octanoic acid and methyl linolenate were detected in ST. For other substances, the methoxybenzene substances with stale odor 1,2,3-trimethoxybenzene and 1,2,4-trimethoxybenzene exhibited the highest content, but with little difference among the three ODT samples. In SL, a unique tea pyrrole (1-ethylpyrrole-2-carbaldehyde) was detected.

OPLS-DA analysis was performed on volatile substances, and 27 differential metabolites were screened according to VIP > 1 (*p* < 0.05) (Figure 3D). Among the 27 differential metabolites screened, alcohols, aldehydes and ketones, and acid and esters exhibited an overall higher content in HA, providing a richer aroma substance basis for HA ODT. The terpene alcohols and aromatic alcohols in alcohols are mostly floral and fruity [16], such as nerol and linalool, which provide pleasant aromas. Terpene alcohols and aromatic alcohols are mainly derived from the metabolism of lipids and hydrolysis of glycoside precursors [17]. HA fixing is characterized by a high temperature and long time, thus favoring the transformation of aroma precursor substances [18]. Meanwhile, aldehydes with citrus fruit aroma, such as citral and decanal, also showed a relatively high content, thus contributing partially to the aroma, while acids and esters with a higher threshold are generally considered to contribute little to the aroma [19]. This may explain why HA ODT was considered to have a strong fruity aroma and good aroma coordination in sensory evaluation. Additionally, ST ODT shows a higher content of limonene and citral with lemon aroma, which was consistent with the research results of baked green tea [20]. Due to obvious lemon aroma and low content of other aroma substances, ST ODT is slightly poor in aroma coordination in the sensory evaluation. Moreover, SL ODT exhibits a higher content of fruity carveol, 1-octanol, and pyrrole with roasted aroma. Pyrroles are mainly produced by Maillard reaction during processing [20], and the soluble protein content is higher in SL-fixed orange peel, which can be further decomposed into amino acids, thereby providing a large amount of precursors for Maillard reaction. Meanwhile, SL ODT is also high in the content of nerol, geraniol, and other substances with pleasant floral and fruity aroma, endowing SL ODT with a special aroma different from HA or ST ODT. 

### 2.4. Effects of Different Fixing Methods on Bioactivities

The antioxidant capacity of different ODT samples was determined using three in vitro antioxidant methods (ABTS, DPPH, and FRAP), and the results are shown in Table 4. For the three ODT samples, HA showed the highest overall value in the three antioxidant capacities. Meanwhile, SL is not significantly different from HA in the antioxidant capacity against ABTS and DPPH, but significantly (*p* < 0.05) lower than HA or ST in the antioxidant capacity against FRAP. Moreover, except for higher antioxidant capacity against FRAP than SL, ST is the lowest in the antioxidant capacity against both DPPH and ABTS. Overall, the three ODT samples varied in the results of the three in vitro antioxidant assays, with the change law of FRAP different from that of DPPH and ABTS, but HA showed consistently relatively high antioxidant capacity in all of the three assays. 

Due to the different principles of the three in vitro antioxidant assay methods, the antioxidant effects of samples cannot be directly compared, so a comprehensive antioxidant evaluation index antioxidant potency composite (APC) value was introduced [21]. After calculation, the APC values of different samples are shown in Table 4. Comparison of the antioxidant indexes among the three samples revealed that HA has an APC index of 100 for both FRAP and DPPH, thus the largest comprehensive APC index of 99.8 and the strongest comprehensive antioxidant capacity, followed by ST (94.15), and SL has the lowest comprehensive APC index of 93.88, thus the weakest comprehensive antioxidant capacity.

The effects of different fixing methods on ODT enzyme activity were investigated by analyzing the half-inhibitory concentrations (IC_50_) against α-glucosidase and α-amylase, and the results are shown in Table 5. The three ODT samples were seen to vary in the inhibition rules of the two enzyme activities. IC_50_ indicates the sample concentration required for the enzyme activity inhibition rate of 50%, and the smaller the IC_50_ is, the stronger the inhibitory effect [22]. Among the three samples, HA has the lowest IC_50_ value for α-glucosidase, followed by ST, and SL has the highest IC_50_ value. However, an opposite pattern was observed for the IC_50_ values of α-amylase, with a significantly (*p* < 0.05) higher IC_50_ value for HA than ST or SL. This indicated that HA has the strongest inhibitory effect on α-glucosidase, while it has the weakest inhibitory effect on α-amylase.

Correlation analysis (Table 6) showed that FRAP inhibition was positively correlated with tea polyphenols content (*p* < 0.01), while DPPH inhibition was significantly (*p* < 0.05) and positively correlated with tea polyphenols and synephrine content and somewhat correlated with flavonoids, polyphenols, and hesperidin. ABTS inhibition was positively correlated with flavonoids, polyphenols, hesperidin, synephrine, and limonin content (*p* < 0.05). For the inhibition of both enzymes, correlation analysis showed that the IC_50_ of α-amylase was positively correlated with the content of tea polyphenols (*p* < 0.01), and the IC_50_ of α-glucosidase was significantly (*p* < 0.01) and positively correlated with the content of soluble protein, flavonoids, and limonin.

### 2.5. Effects of Different Temperatures of HA Fixing on Sensory Quality

The above analysis indicated HA fixing is obviously better than ST or SL fixing in improving ODT quality, so this experiment further optimized the temperatures of HA fixing, and the sensory evaluation results are shown in Table 7. The ODT sensory scores were seen to increase gradually with the increase in fixing temperature, and the ODT samples fixed at different temperatures have a bright red soup, high fruity aroma, and good coordination, with little difference between different ODT samples in aroma and soup color but some difference in appearance and taste.

In terms of appearance/peel color, with the increase in fixing temperature, the orange peel color gradually turned from green to yellowish brown, probably because the increased temperature promoted the formation of melanin in the peel, thereby making the peel gradually darkened and increasing the non-uniformity of peel color [23], leading to a gradual decrease in appearance score. Additionally, all samples showed the characteristics of red and bright in terms of soup color and the characteristics of “fruity aroma and coordinate” in terms of aroma, indicating no difference in the soup color and aroma of ODT samples fixed at different temperatures.

The taste factor scores of ODT samples fixed at different temperatures are shown in Figure 4. It was shown that with the increase in fixing temperature, the scores of sourness and sweetness showed no obvious change, in contrast to a gradual increase in the scores of bitterness, astringency, and coordination. In terms of bitterness and astringency, both scores increased with the increase in fixing temperature, probably because the thermal action induced by the increased temperature promotes the degradation and oxidation of phenols [24], facilitating their complexation with alkaloids and proteins to form water-insoluble macromolecular compounds, resulting in a decrease in the content of phenols. Meanwhile, higher temperatures may also cause more degradation of limonin [25], resulting in a lower level of bitter substance in ODT. With the decrease in bitterness and astringency, the coordination of tea soup gradually increased, and the overall score increased.

### 2.6. Effects of Different Temperatures of HA Fixing on Chemical Composition

The main components of the orange peel fixed at different temperatures are shown in Figure 5A,B. With the increase in temperature, the orange peel decreased significantly (*p* < 0.05) in the content of polysaccharides, flavonoids, hesperidin, synephrine, and limonin, with a plateau for polysaccharides, polyphenols and hesperidin at 85~95 °C. In addition, polyphenols and soluble proteins also showed a slight downtrend. As mentioned above, the increase in temperature caused more degradation and transformation of temperature-sensitive proteins and polyphenols in the peel, resulting in a decrease in their content. The changes of polysaccharides and synephrine may be related to their stability induced by high temperature, resulting in alterations in their structures [26]. However, limonin is responsible for the obvious bitter and astringent taste in the peel. With the increase in fixing temperature, both limonin content and bitterness decreased, agreeing with the sensory evaluation results. The main components of tea at different fixing temperatures are shown in Figure 5C,D, and the main components in the tea were seen to remain almost unchanged with the increase in fixing temperature, probably because ODT is made by filling the orange peel with dry tea, allowing the tea leaves to retain their main components during the reprocessing process.

Comprehensive analysis of the differences in the quality of ODTs fixed at different temperatures showed that the higher the fixing temperature, the lower the content of polyphenols, polysaccharides, limonin, and other substances in ODT, coupled with an increase in sensory evaluation scores. Therefore, a fixing temperature in the range of 85~95 °C can be considered as a suitable temperature for fixing ODT, not only retaining more peel and tea inclusions, but also achieving better sensory quality.

## 3. Discussion

In the traditional citrus-tea production process, fixation is based on outdoor sunlight, but the quality is unstable due to weather changes. In recent years, the new fixing methods such as steam and hot air fixing have been developed, but few studies have been performed on the use of these methods in citrus-tea production.

From the results of this experiment, the fixing method mainly affected the appearance and taste, especially on the taste. The effect on taste was reflected in sourness, bitterness, astringency, and coordination. This phenomenon is also present in the drying of coffee beans and tea. Dong et al. found a more sour taste for sun-dried coffee beans [27]. ST-fixed green tea is generally heavier in bitterness and astringency than HA-fixed green tea [28], probably due to its high treatment temperature and short treatment time, resulting in less conversion and degradation of bitter substances (such as phenols) in the sample, and thus heavier bitterness and astringency. Due to lower temperature, SL fixing was reported to retain more bitter substances such as phenols and limonin in the orange peel, leading to obvious astringency in ODT [29].

The different fixing methods had significant effects on the main physicochemical components of the peel, but less so on the tea. Our experiments revealed that the soluble protein, polyphenols, flavonoids, hesperidin, synephrine, and limonin contents of sunlight-fixed peel were significantly higher than HA and ST, probably due to sunlight-fixed relatively lower temperature, slowing down the degradation of heat-sensitive substances, such as soluble protein, and favoring the retention of functional substances in the peel [30]. Additionally, polyphenols are easy to react with proteins at high temperature, thus reducing the content of phenolics in HA- or ST-fixed peel. For limonin, several previous studies have found that limonin content is negatively correlated with temperature [25]. Of the three fixing methods, SL had the lowest temperature, thus the highest limonin content and a more bitter taste for the peel in sensory evaluation. The reason for the smaller effect of different fixing methods of the tea in ODT may be since orange dark tea is filled with dark tea, which is a deeply fermented tea, and the tea polyphenols have undergone deep oxidation and polymerization. The three fixing methods varied in thermal conduction mechanism and reaction environment, as well as chemical reaction, which contributed jointly to the difference in aroma type and intensity [18].

This study reveals for the first time the effect of the fixing method on the antioxidant activity of orange dark tea and the inhibition of α-glucosidase and α-amylase activities. Citrus fruits are natural antioxidants with good antioxidant capacity. In the two antioxidant determination methods of DPPH and ABTS, the antioxidant activity of SL is stronger and higher than that of ST, which is consistent with the content of polyphenols and flavonoids in the peel. Therefore, polyphenols and flavonoids can be assumed to be an important source of antioxidant capacity of citrus tea. The research results of Chen et al. also indicated total phenols and flavonoids in orange peel as the basis of antioxidant capacity [31]. Despite their similar antioxidant capacity against DPPH and ABTS, HA and SL are different in the content of polyphenols. A possible explanation is their little difference in the content of synephrine. FRAP measurement results revealed that HA has the highest antioxidant capacity against FRAP. Despite a lower content of polyphenols in HA peel than in SL peel, HA is higher than SL in the content of tea polyphenols. Meanwhile, ODT is mainly made by filling dry tea leaves into the orange peel, thus endowing HA of a higher tea polyphenol content with stronger antioxidant capacity against FRAP and stronger comprehensive antioxidant capacity. Previous studies have shown that polyphenols can significantly affect the inhibitory activity against α-amylase [32]. Therefore, HA ODT has a higher IC_50_ of α-amylase and thus is lower than ST or SL ODT in the inhibitory effect on α-amylase. For α-glucosidase, flavonoids have a strong inhibitory effect, so bitter tea with a high flavonoid content can be used to prevent hyperglycemia, and soluble protein may also have a certain inhibitory effect [33]. Therefore, the higher content of these substances in HA ODT enabled it to have a better inhibitory effect on α-glucosidase, also indicating its better hypoglycemic function [34].

## 4. Materials and Methods

### 4.1. Materials and Reagents

The peach leaf orange was picked from Longmaxi Village, Quyuan Town, Zigui County (Hubei, China) on 3 September 2019; the Qingzhuan tea (loose tea) was provided by Zigui Yihong Tea Co., Ltd. (Hubei, China) 

Folin phenol, ninhydrin, methanol, anhydrous ethanol, anthrone, sulfuric acid, potassium dihydrogen phosphate, disodium hydrogen phosphate, sodium carbonate, oxalic acid, ethyl acetate, n-butanol, stannous chloride, Coomassie brilliant blue G250, sodium chloride, and phosphoric acid were all of analytical grade and purchased from China Pharmaceutical (Group) Shanghai Chemical Reagent Company (Shanghai, China). Bovine serum albumin BSA, α-glucosidase, α-glucoside, rutin, limonin, synephrine, hesperidin, and cyclohexanone standard products were purchased from Shanghai Yuanye Biotechnology Co., Ltd. (Shanghai, China). ABTS, FRAP, and DPPH antioxidant kits were purchased from Suzhou Keming Biotechnology Co., Ltd. (Suzhou, Jiangsu, China). α-amylase activity detection reagent cartridges were purchased from Solarbio Co., Ltd. (Beijing, China).

### 4.2. Sample Preparation

Three types of orange dark tea were prepared as shown in Figure 6. Specifically, the orange was rinsed with clean water, followed by hollowing out the pulp, cleaning, drying and filling tea leaves into the husk to three quarters, and fixing them together separately by hot air (HA), steam (ST), or sunlight (SL):

**Hot air** (**HA**) **fixing:** 6CTH-6.0 box-type drying machine (Zhejiang Green Peak Machinery Co., Ltd., Zhejiang, China) was used to fix ODT under the conditions of raising the temperature to 50 °C for 10 min, then up to 70 °C and holding for 20 min, up to 85 °C and holding for 10 min, up to 90 °C and holding for 10 min, and finally down to 50 °C and holding for 20 min. 

**Steam** (**ST**) **fixing:** A steamer was used to fix ODT by maintaining the temperature at 100 °C for 5 min. 

**Sunlight** (**SL**) **fixing:** ODT was fixed by exposure to sunlight on bamboo strips for 5 h at 40~50 °C.

**Different HA fixing temperatures:** ODT was fixed separately for 20 min at 70 °C, 75 °C, 80 °C, 85 °C, 90 °C, 95 °C, and 100 °C using a 6CTH-6.0 box-type drying machine (Zhejiang Green Peak Machinery Co., Ltd., Zhejiang, China). 

After fixing treatment, all three groups of ODT samples were dried with the 6CTH-6.0 box-type drying machine first at 45 °C for 2 h, followed by heating at 50 °C for 6 h and 80 °C for 10 h, then cooling to 65 °C for 2 h, and finally heating at 75 °C for 15 min.

### 4.3. Sensory Evaluation

Tea sensory evaluation was performed as previously reported with appropriate modifications [35]. Briefly, 3 g of the mixed tea leaves and 1.2 g of orange peel were collected from each sample and placed in a 150 mL evaluation cup, followed by filling the cup with boiling water, covering it, and steeping for 5 min. Next, the tea soup was poured into an evaluation tea bowl at the same speed and in the sequence of brewing for evaluation of its color, aroma, and taste, with aroma evaluated in terms of concentration, fruit aroma, pleasantness, and coordination, with taste in terms of acidity, sweetness, body, bitterness, astringency, and harmony. The data for the experts were collected and stored in accordance with Huazhong Agricultural University Human Ethics application ID Number: HZAUHU-2020-0018 (Appendix A).

### 4.4. Chemical Composition Determination

Polyphenols in tea and orange peel [36]: Each sample powder (0.2 g) was mixed with 10 mL of 70% methanol, followed by extraction in a water bath at 70 °C, and determining the content of polyphenols using the Folin phenol colorimetric method. 

Free amino acids and soluble sugars in tea [37]: Each sample (0.5 g) was added with 50 mL of boiling water for extraction in a boiling water bath, and, after filtering the tea soup, the ninhydrin colorimetry and the anthrone-sulfuric acid colorimetry were used to determine the content of free amino acids and soluble sugars, respectively.

Determination of orange peel polysaccharide [38]: For each sample (0.2 g), 40 mL of 80% ethanol was added and extracted by reflux in a 95 °C water bath, followed by adding 100 mL of distilled water for extraction in a boiling water bath, then filtering the tea soup and determining the orange peel polysaccharide using the anthrone sulfuric acid method. 

Determination of theaflavins, thearubigins, and theabrownins in tea [35]: For each sample powder (3 g), 125 mL of boiling water was added, followed by extraction in a boiling water bath, then suction, filtering the tea soup, and determining the content of theaflavins, thearubigins, and theabrownins using the system detection method. 

Flavonoids in orange peel [39]: Each sample (0.2 g) was exposed to ultrasonic extraction with absolute ethanol for 30 min, followed by filtration and determining the content of flavonoids using the aluminum nitrate method.

Soluble protein in orange peel [40]: Each sample (0.2 g) was extracted with boiling water at 100 °C for 10 min, followed by centrifugation at 3000 rpm for 10 min to collect the supernatant for determination of soluble protein using the Coomassie brilliant blue method.

Determination of hesperidin, synephrine, and limonin in orange peel: Simultaneous determination by HPLC [41]. Extraction preparation: For each ground orange peel powder (0.1 g), 10 mL of methanol was added, followed by ultrasonic extraction for 30 min, filtration, dilution to 10 mL, and passing 1 mL diluted solution through a 0.22 μm filter membrane for HPLC determination of hesperidin, synephrine, and limonin as described below. 

HPLC conditions: an Agilent ZORBAX SB-C 18 chromatographic column (250 mm × 4.6 mm × 5 μm, Agilent, Santa Clara, CA, USA) was used: flow rate, 1 mL·min-1; column temperature, 35 °C; injection volume, 5 μL; detection wavelength, 210 nm and 283 nm; mobile phase A, aqueous phosphoric acid solution with pH = 3.7; mobile phase B, methanol: acetonitrile = 1:1, with the gradient elution as 0~5 min, A:B = 100:0; 5~10 min, A:B = 95:5; 10~20 min, A:B = 75:25; 20~25 min, A:B = 50:50; 25~30 min, A:B = 25:75; 30~40 min, A:B = 5:95; 5~10 min, A:B = 95:5.

### 4.5. Determination of Volatile Components in Citrus Tea

Extraction of aroma compounds by headspace solid phase microextraction (HS-SPME) [22]: Briefly, DVB/CAR/PDMS extraction fiber was inserted into the GC injection port and aged at 250 °C for 30 min. Next, each ODT powder (1 g) was put into a 20 mL headspace bottle, followed by adding 5 mL of boiling saturated NaCl solution and 1 mL of cyclohexanone internal standard, closing the bottle immediately, and placing the headspace vial in a water bath at 60 °C for 1 h. The DSQ-II gas-mass spectrometer (Thermo Fisher Scientific, Waltham, MA, USA) was used for chromatographic analysis under the following conditions: the chromatographic column, DB-5MS (30 mm × 0.25 mm × 0.22 μm); inlet temperature, 230 °C; carrier gas, high-purity helium, with purity ≥99.99%; column flow rate, 1.0 mL/min. The heating program was as follows: the initial temperature was 45 °C, up to 80 °C at 7 °C/min without holding, up to 90 °C at 2 °C/min and holding for 2 min, up to 100 °C at 3 °C/min and holding for 2 min, up to 130 °C at 3 °C/min for 2 min, up to 150 °C at 3 °C/min, finally up to 230 °C at 10 °C/min and holding for 5 min. The oven temperature was set at 40 °C, and the sample was injected splitless. Mass spectrometry conditions were: ion source EI, electron energy 70 eV, ion source temperature 230 °C, and mass scanning range 30–500 m/z.

### 4.6. Antioxidant Activity and Enzyme Activity Inhibition Assay 

The ODT antioxidant capacity was determined using FRAP, ABTS, and DPPH kits as instructed by the manufacturer (Suzhou Keming Biotechnology Co., Ltd.). Briefly, each sample powder (0.1 g) was weighed into a 10 mL centrifuge tube, followed by adding 10 mL of boiling distilled water, incubation in a boiling water bath for 10 min, and centrifugation for 5 min to collect the supernatant for testing as described below.

APC is calculated by the equation according to the reference [21]:APC = (Index _DPPH 1_ + Index _ABTS_ + Index _FRAP_)/3 

FRAP: The mixed solution was prepared as instructed by the manufacturer, followed by adding 190 μL of the mixed solution to the 96-microwell plate reaction system, and then adding 10 μL of blank and different samples for respective tests. After standing for 20 min, the absorbance at 593 nm was measured, and the inhibitory activity was determined by Equation (1):ΔA = A_assay_ − A_blank_,(1)

ABTS: The reagents were used to prepare the mixed solution as instructed by the manufacturer. Next, 190 μL of the mixed solution was added to the 96-well plate reaction system, followed by adding 10 μL of blank and different samples and standing at room temperature for 10 min. Finally, the absorbance was measured at 734 nm and the inhibitory activity was estimated by Equation (2): ΔA = A_blank1_ − A_assay1_,(2)

DPPH: The reagents were used to prepare the mixed solution as instructed by the manufacturer, followed by adding 380 μL working solution to 500 μL test tube, then adding 20 μL blank and different samples. After reaction at room temperature for 20 min in the dark, 200 μL was collected into a 96-well plate to measure the absorbance at 515 nm and determine the inhibitory activity by Equation (3):ΔA = A_blank2_ − A_assay2_,(3)

α-amylase activity inhibition assay was performed as instructed by the kit manufacturer (Beijing Solarbio Technology Co., Ltd.). The control tube, reaction tube, inhibition tube, background tube, and reagents were prepared separately as previously reported [22]:

Control tube: 550 μL of distilled water, 0 μL of α-amylase solution (3.3 μg/mL), and 0 μL of tea soup.

Reaction tube: 0 μL of distilled water, 200 μL of α-amylase solution (3.3 μg/mL), and 0 μL of tea soup. 

Inhibition tube: 150 μL of distilled water, 200 μL of α-amylase solution (3.3 μg/mL), and 200 μL of tea soup.

Background tube: 350 μL of distilled water, 0 μL of α-amylase solution (3.3 μg/mL), and 200 μL of tea soup.

Next, all the tubes were placed in a 70 °C water bath for 15 min, followed by cooling in ice water for 3 min, then adding 150 μL of Reagent 2 to the reaction tube and the inhibition tube, incubation in a 40 °C water bath for 5 min, then adding 150 μL of Reagent 1 to each of the four tubes, further incubation in a 100 °C water bath for 10 min, cooling in ice water for 5 min, and measuring OD control, OD reaction, OD inhibition, and OD background at 540 nm.

The inhibition rate of α-amylase activity was calculated by Equation (4):(4)$$\alpha \text{-amylase}\;\mathrm{activity}\;\mathrm{inhibition}\;\mathrm{rate}=\left(1-\frac{{OD}_{inhibition\;}-OD_{background}}{OD_{reaction}-OD_{control}}\right)\times 100,$$

The α-glucosidase inhibitory activity was measured as previously reported [22]. Briefly, 40 μL of α-glucosidase solution at 1 unit/mL concentration was added to the 96-microwell plate reaction system, followed by adding 40 μL of each sample at different concentrations, reaction at 37 °C for 10 min, then adding 40 μL of 2.5 mmol/mL pNPG solution, incubation at constant temperature (37 °C) for 30 min, and terminating the reaction by adding 120 μL of sodium carbonate solution at 0.2 mol/L concentration. Finally, the absorbance value at 405 nm was measured, and IC_50_ was calculated.

### 4.7. Data Analysis

Data were presented as mean ± SD with 3 replicates. SPSS Statistics 26 software was used for statistical analysis, and differences between groups were analyzed using one-way ANOVA, followed by least significant difference (LSD) multiple comparisons, with *p* < 0.05 considered significantly different. Radar, histogram, and line graph were drawn using Origin 2021 software, and heat map was created using TBtools. SIMCA 14.1 software was used for principal component analysis (PCA) and graphing.

## 5. Conclusions

This study compared the quality of orange dark tea fixed by hot air, steam, and traditional sunlight, and the hot air-fixed orange dark tea was shown to be in good harmony with the aroma of the fruit and the tea, with a mellow taste and high sensory quality. The hot-air-fixed orange dark tea has strong comprehensive antioxidant capacity and strong inhibitory activity against α-glucosidase. Temperature optimization revealed that hot air fixing at 85~95 °C can not only retain more peel and tea inclusions, but also achieve better sensory quality. Overall, this study provides a theoretical basis for the production and processing of citrus tea and helps to optimize the processing technology of citrus tea to improve its quality.

## Figures and Tables

**Figure 1 molecules-28-01079-f001:**
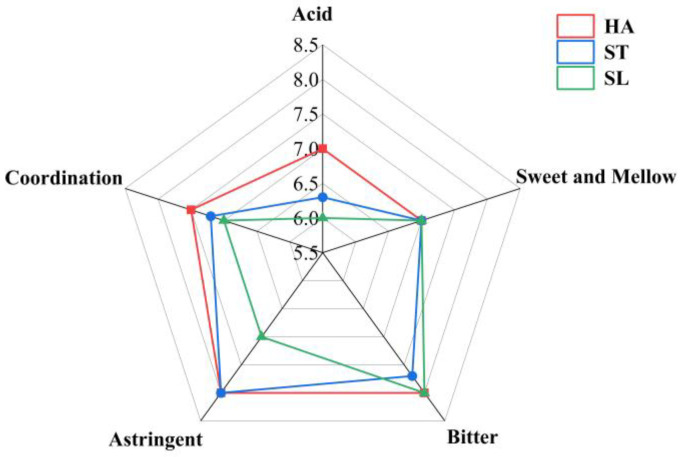
Radar chart of citrus tea taste factor scores for different fixing methods.

**Figure 2 molecules-28-01079-f002:**
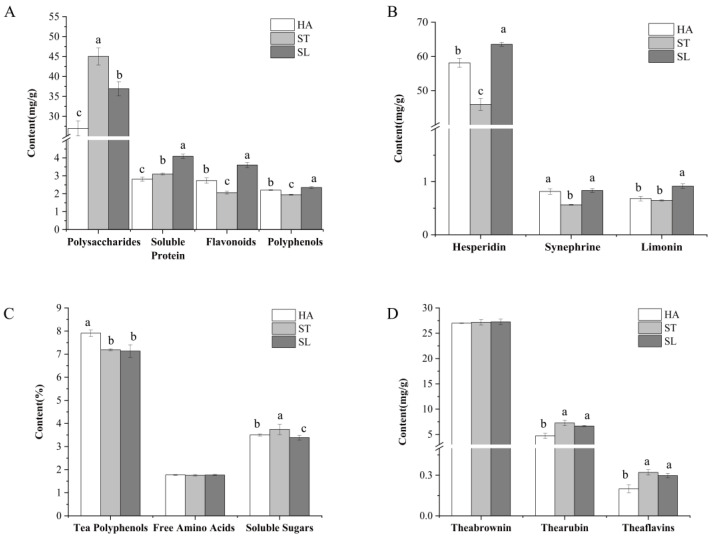
Effects of different fixing methods on main biochemical compositions of orange dark tea. (**A**,**B**) Effects of different fixing methods on the main chemical components in the peel. (**C**,**D**) Effects of different fixing methods on the main chemical components in the tea. HA, hot air fixing; ST, steam fixing; SL, sunlight fixing. Different small letters indicate significant difference at *p* < 0.05.

**Figure 3 molecules-28-01079-f003:**
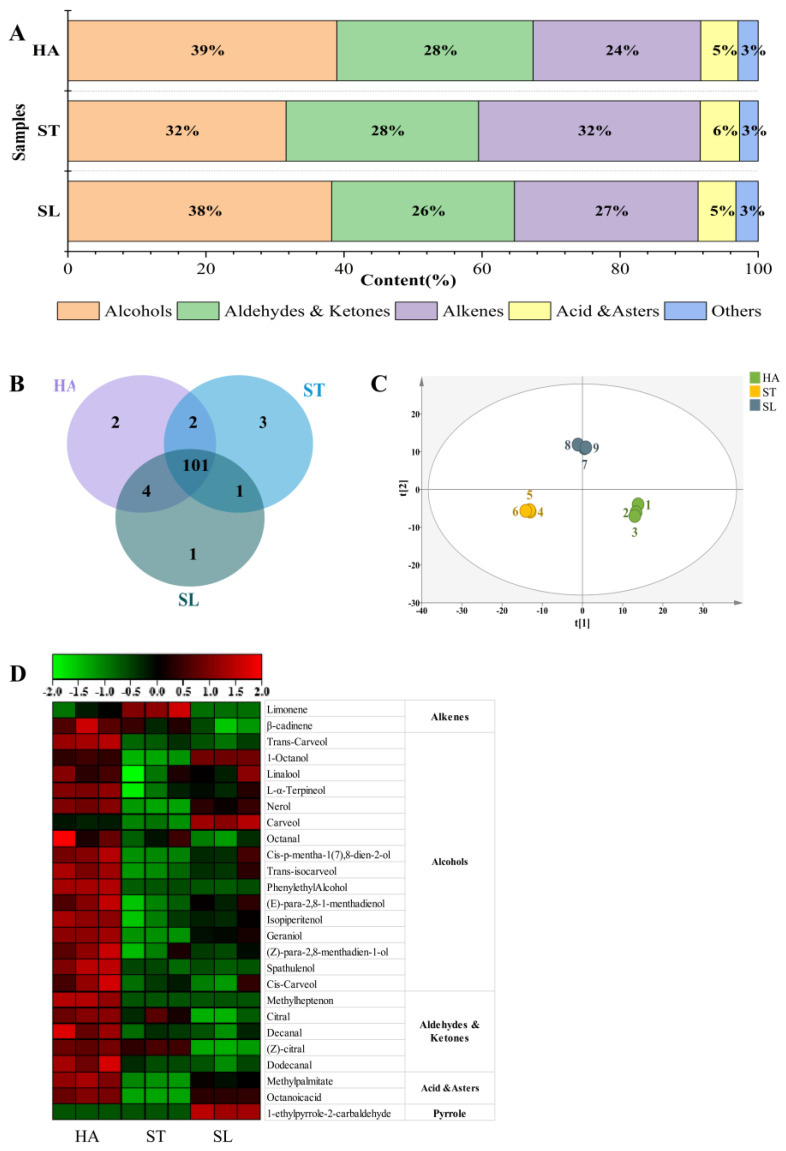
Effects of different fixing methods on the content of volatile compounds in orange dark tea. (**A**) Main volatile components in differently fixed ODTs; (**B**) Venn diagram of volatile compounds in differently fixed ODTs; (**C**) OPLS-DA score plot (R^2^Y = 0.996, Q^2^ = 0.966). (**D**) Heat-map for the main volatile compounds in differently fixed dark orange teas.

**Figure 4 molecules-28-01079-f004:**
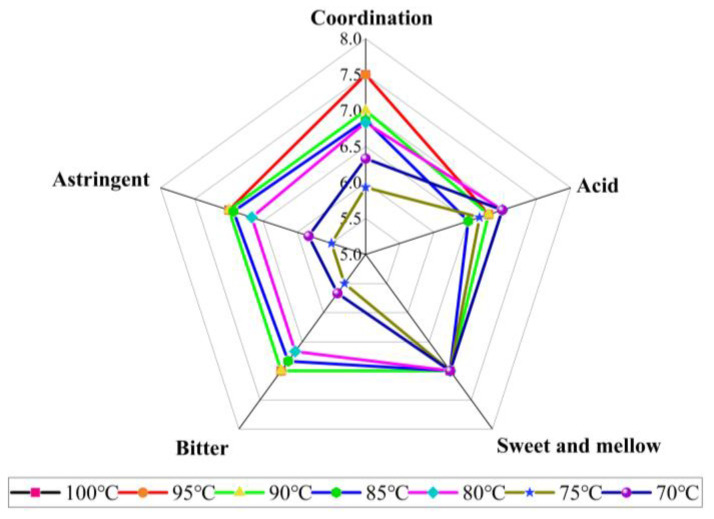
Radar chart of ODT flavor factor scores at different fixing temperatures.

**Figure 5 molecules-28-01079-f005:**
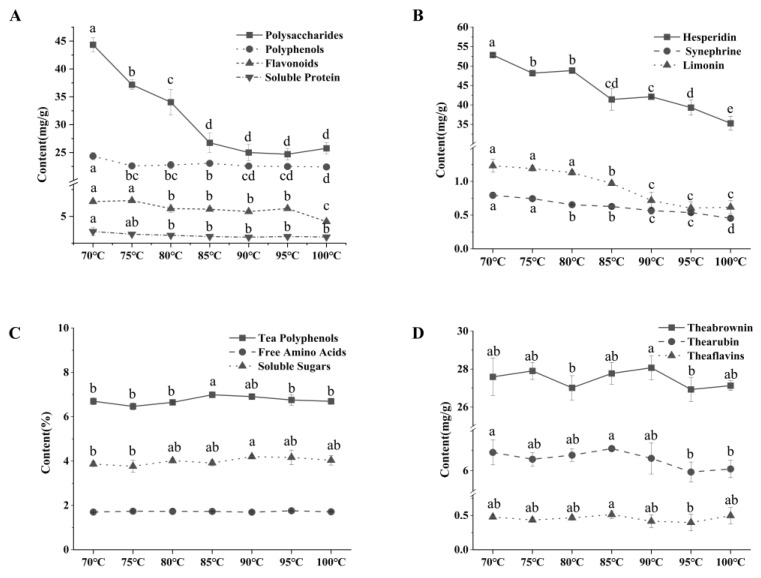
Physicochemical components of orange peel and tea at different fixed temperatures. (**A**) Main physicochemical components of orange peel. (**B**) Orange peel active substance. (**C**) The main physicochemical components of tea. (**D**) Tea pigments. (Different lowercase letters on the figure indicate significant difference at *p* < 0.05).

**Figure 6 molecules-28-01079-f006:**
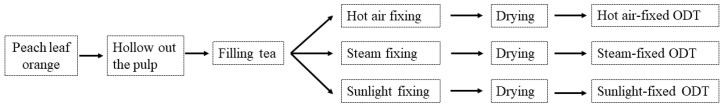
Flow diagram for processing different types of orange dark tea.

**Table 1 molecules-28-01079-t001:** Sensory evaluation of ODT samples with different fixing methods.

Sample	Appearance (10 Points)	Soup Color (10 Points)	Aroma (30 Points)		Taste (50 Points)					Total Score
			Aroma	Coordination	Sour	Sweet	Bitter	Astringency	Coordination	
HA	6.20 ± 0.30 ab	9.10 ± 0.10	15.30 ± 0.10	9.0 ± 0.00	7.00 ± 0.00 a	7.00 ± 0.00	8.00 ± 0.00 a	8.00 ± 0.00 a	7.50 ± 0.00 a	77.00 ± 0.50 a
ST	5.80 ± 0.30 b	9.00 ± 0.00	15.00 ± 0.00	9.0 ± 0.00	6.50 ± 0.60 b	7.00 ± 0.00	7.70 ± 0.30 a	7.80 ± 0.20 a	7.20 ± 0.30 b	75.00 ± 0.90 b
SL	6.50 ± 0.30 a	9.10 ± 0.10	15.00 ± 0.10	9.0 ± 0.00	6.00 ± 0.00 b	7.00 ± 0.00	7.80 ± 0.00 ab	7.00 ± 0.00 b	7.20 ± 0.00 b	74.60 ± 0.40 b

**Note:** Different lowercase letters in each column indicate significant difference at *p* < 0.05.

**Table 2 molecules-28-01079-t002:** Analysis of volatile components in orange dark tea with different fixing methods (μg/g).

RT	RI	Compound Name	Odor Description	HA	ST	SL
		**Alcohols**				
6.35	969	1-Heptanol	sweet, woody	0.57 ± 0.08	0.36 ± 0.01	-
9.27	1070	1-Octanol	waxy, green, fruity	24.70 ± 0.72 b	7.52 ± 1.30 c	29.35 ± 0.11 a
9.83	1086	Terpinolene	fresh, woody, floral	4.24 ± 0.19 a	3.65 ± 0.22 b	3.11 ± 0.16 c
10.51	1102	Linalool	floral, sweet	173.54 ± 5.41	150.14 ± 17.27	168.25 ± 10.53
10.87	1113	Phenylethyl alcohol	floral, sweet, rosy	13.35 ± 0.28 a	4.33 ± 0.34 b	4.36 ± 0.26 b
11.31	1123	(E)-para-2,8-1-menthadienol	fresh, minty	23.5 ± 1.62 a	15.83 ± 1.45 c	19.91 ± 1.06 b
11.99	1137	(Z)-para-2,8-menthadien-1-ol	null	26.14 ± 1.51 a	19.57 ± 3.01 b	20.89 ± 0.88 b
12.55	1150	Isopulegol	minty, cooling, woody	1.20 ± 0.14 a	1.33 ± 0.10 a	0.90 ± 0.12 b
13.65	1172	1-Nonanol	fresh, fatty, floral	8.37 ± 0.22 a	4.07 ± 0.07 c	7.40 ± 0.45 b
13.82	1176	Linalool oxide (pyranoid)	floral, honey	4.86 ± 0.40 a	3.39 ± 0.09 b	2.65 ± 0.20 c
14.09	1181	4-Terpineol	pepper, woody, musty	26.57 ± 1.45	22.59 ± 1.48	24.09 ± 2.66
14.42	1188	Trans-isocarveol	null	32.48 ± 1.08 a	20.22 ± 1.14 c	25.07 ± 2.33 b
14.64	1191	2,6-dimethyl-3,7-Octadiene-2,6-diol,	null	0.89 ± 0.10 b	0.73 ± 0.02 c	1.41 ± 0.08 a
14.9	1196	L-α-Terpineol	floral, terpenic	76.31 ± 0.80 a	55.44 ± 8.19 c	65.3 ± 3.11 b
14.96	1197	Dihydrocarveol	minty, herbal	-	2.28 ± 0.19	-
15.07	1198	Isopiperitenol	null	17.18 ± 0.41 a	9.61 ± 1.94 c	12.47 ± 0.65 b
15.96	1216	Carveol	green, weedy, herbal	12.30 ± 0.24 b	8.59 ± 0.55 c	19.03 ± 0.79 a
16.06	1218	Trans-carveol	caraway, spearmint	48.54 ± 1.36 a	26.21 ± 2.55 b	25.65 ± 2.50 b
16.37	1224	Nerol	sweet, citrus, green	22.95 ± 0.64 a	7.50 ± 0.32 c	18.15 ± 1.23 b
16.62	1229	Cis-p-mentha-1(7),8-dien-2-ol	null	34.75 ± 1.52 a	21.96 ± 0.32 c	27.94 ± 2.93 b
16.76	1231	Cis-carveol	caraway	21.66 ± 1.47 a	17.38 ± 0.87 b	17.12 ± 2.27 b
17.82	1251	Geraniol	floral, rosy, waxy	15.70 ± 0.26 a	8.60 ± 0.04 c	12.14 ± 0.48 b
18	1254	2-Methoxybenzylalcohol	anisic	1.13 ± 0.05 b	3.08 ± 0.13 a	2.48 ± 0.41 a
20.02	1287	2-(4-Methylenecyclohexyl)-2-propen-1-ol	null	18.22 ± 0.98 a	16.11 ± 1.09 b	15.91 ± 1.23 b
20.5	1295	Perilla alcohol	woody, spicy, floral	7.40 ± 0.57 a	6.34 ± 0.23 b	6.74 ± 0.26 a
32.97	1537	α-elemol	green, woody, spicy	9.01 ± 0.67 b	8.92 ± 0.81 b	10.85 ± 0.42 a
33.75	1553	Nerolidol	green, floral, woody	4.13 ± 0.50 b	3.53 ± 0.27 c	4.86 ± 0.43 a
34.19	1562	Spathulenol	earthy, herbal, fruity	8.04 ± 0.65 a	3.41 ± 0.39 b	3.35 ± 0.15 b
37.04	1635	Cubenol	spicy, herbal, green tea	2.79 ± 0.12 a	1.42 ± 0.05 b	0.94 ± 0.13 c
37.38	1646	β-Eudesmol	woody, green	7.00 ± 0.62 a	5.80 ± 0.12 b	5.65 ± 0.47 b
		**Aldehydes and Ketones**				
4.16	854	(E)-2-Hexenal	green, banana, fatty	0.38 ± 0.06	-	1.01 ± 0.03
6.03	956	(E)-2-Heptenal	green, sweet, fruity	0.55 ± 0.04	-	0.23 ± 0.01
6.17	961	Benzaldehyde	fruity, cherry, oily	1.23 ± 0.16 a	0.20 ± 0.03 c	0.57 ± 0.03 b
6.67	983	Methylhepten	citrus, green, musty	8.57 ± 0.67	-	-
7.18	1003	Octanal	waxy, fatty, citrus	58.28 ± 5.16 a	51.67 ± 3.58 ab	47.46 ± 2.67 b
8.38	1044	Benzeneacetaldehyde	honey, sweet, floral	2.00 ± 0.17 a	0.64 ± 0.07 c	1.00 ± 0.05 b
10.61	1106	Nonanal	waxy, aldehydic, rose	23.12 ± 2.18 a	20.69 ± 0.59 a	18.88 ± 0.94 b
12.7	1153	Citronellal	floral, green, rosy	13.71 ± 0.73	12.46 ± 0.73	13.57 ± 0.79
13.06	1161	(E)-2-Nonenal	green, soapy, cucumber	0.84 ± 0.05 b	0.73 ± 0.09 b	1.02 ± 0.07 a
15.29	1207	Decanal	waxy, fatty, citrus	95.24 ± 4.32 a	78.84 ± 2.38 b	77.62 ± 3.93 b
15.67	1211	Berbenone	camphor, menthol, celery	2.91 ± 0.14	2.84 ± 0.13	2.77 ± 0.14
17.12	1238	(Z)-citral	sweet, citral, lemon peel	28.15 ± 0.46 a	26.23 ± 0.59 a	16.54 ± 0.46 b
17.42	1243	(+)-carvone	spice, mint, caraway	36.63 ± 2.27	33.11 ± 1.66	35.75 ± 1.67
18.5	1263	Trans-2-Decenal	waxy, fatty, earthy	3.33 ± 0.12 a	1.79 ± 0.21 b	2.83 ± 0.41 a
18.89	1269	Citral	citrus, juicy, green	50.95 ± 0.96 a	44.63 ± 4.03 b	33.9 ± 3.15 c
19.16	1274	Perillaldehyde	fresh, green, herbal	35.96 ± 2.06	32.57 ± 1.43	34.04 ± 3.61
20.28	1291	2-Undecanone	waxy, fruity, creamy	1.63 ± 0.02 a	1.29 ± 0.15 b	1.48 ± 0.12 ab
21.12	1306	Undecanal	waxy, soapy, floral	19.03 ± 0.60 a	14.83 ± 0.93 b	15.32 ± 1.40 b
22.44	1333	Piperitenone	minty, phenolic	4.7 ± 0.16 b	6.73 ± 0.29 a	6.19 ± 0.41 a
26.27	1404	Dodecanal	soapy, waxy, citrus	34.65 ± 1.78 a	27.01 ± 0.64 b	25.87 ± 1.26 b
26.64	1412	α-Ionone	sweet, woody, floral	3.22 ± 0.15 a	2.44 ± 0.09 b	2.32 ± 0.19 b
28	1439	Nerylacetone	fatty, metallic	10.72 ± 0.31 a	7.68 ± 0.27 b	10.56 ± 0.42 a
29.46	1467	β-ionone	woody, floral, berry	12.04 ± 0.41 a	11.73 ± 0.79 ab	10.84 ± 0.55 b
29.61	1469	β-Ionone epoxide	fruity, sweet, berry	2.41 ± 0.13 a	2.13 ± 0.12 b	2.32 ± 0.01 a
33.36	1545	4-Isopropyl-2-methyl-2-cyclohexen-1-one	null	3.45 ± 0.49 a	3.35 ± 0.17 ab	3.03 ± 0.16 b
36.17	1604	Tetradecanal	fatty, lactonic, coconut	1.70 ± 0.14 a	1.32 ± 0.06 b	0.85 ± 0.10 c
38.5	1685	β-sinensal	orange, sweet, fresh	11.73 ± 0.74 a	9.89 ± 0.41 c	10.47 ± 0.40 b
39.69	1741	α-Sinensal	citrus, juicy, waxy	2.37 ± 0.22 b	5.89 ± 0.30 a	5.83 ± 0.47 a
41.27	1837	Hexahydrofarnesyl acetone	oily, herbal, jasmin	2.84 ± 0.25 c	4.93 ± 0.18 a	3.02 ± 0.23 b
42.22	1904	Farnesyl acetone	flower, ether	0.33 ± 0.01	0.35 ± 0.04	-
		**Alkenes**				
5.58	934	α-Pinene	fresh, camphor, sweet	1.98 ± 0.01 b	1.45 ± 0.11 c	4.79 ± 0.08 a
6.47	973	β-Pinene	fresh, piney, woody	1.94 ± 0.15 c	5.12 ± 0.16 a	3.43 ± 0.2 b
6.82	989	β-Myrcene	woody, vegetative, citrus	10.81 ± 0.61 b	12.94 ± 2.13 a	12.73 ± 0.74 a
7.61	1018	α-Terpinene	terpy, woody, piney	2.28 ± 0.13 b	1.86 ± 0.31 b	3.00 ± 0.19 a
8.06	1033	Limonene	citrus, herbal, camphor	213.4 ± 16.56 b	271.32 ± 11.51 a	196.02 ± 0.16 a
8.46	1046	Trans-β-Ocimene	fruity, floral	-	2.44 ± 0.34	-
8.87	1059	γ-Terpinene	terpy, citrus, oily	5.10 ± 0.43 c	6.18 ± 0.17 b	7.71 ± 0.65 a
10.88	1113	1,3,8-p-Menthatriene	oily, terpy, camphorous	4.37 ± 0.48 b	5.07 ± 0.19 a	5.32 ± 0.33 a
11.66	1137	4-Acetyl-1-methylcyclohexene	null	1.20 ± 0.21 c	1.80 ± 0.08 b	2.04 ± 0.09 a
20.75	1298	Dipentene dioxide	mentholic	2.04 ± 0.08 a	1.59 ± 0.06 b	1.16 ± 0.09 c
22.99	1344	(-)-α-Cubebene	herbal, waxy	17.46 ± 0.39	18.12 ± 1.42	16.90 ± 0.67
24.36	1370	α-Copaene	woody, spicy, honey	18.52 ± 2.17	18.42 ± 0.78	17.38 ± 0.61
25	1381	β-cubebene	citrus, fruity, radish	5.06 ± 0.15 a	3.84 ± 0.23 b	4.83 ± 0.60 a
25.08	1383	β-Elemen	sweet	9.65 ± 0.88	10.40 ± 0.64	9.69 ± 0.17
26.42	1406	β-Longipinene	null	5.34 ± 0.45	5.42 ± 0.30	6.02 ± 0.31
26.5	1409	Alloocimenal	null	4.01 ± 0.18 a	3.75 ± 0.17 a	3.41 ± 0.2 b
26.92	1417	(-)-β-Copaene	null	8.84 ± 0.33	9.38 ± 0.73	9.11 ± 0.59
27.82	1435	β-copaene	null	2.11 ± 0.11	1.99 ± 0.04	1.92 ± 0.16
28.1	1441	α-Humulene	woody	5.20 ± 0.59 a	4.97 ± 0.25 a	5.26 ± 0.27 b
28.27	1444	Cis-β-Farnesene	vitrus, green	8.62 ± 0.59	8.27 ± 0.43	8.90 ± 0.17
29.04	1459	γ-Gurjunene	null	3.97 ± 0.13 a	3.52 ± 0.38 ab	3.24 ± 0.11 b
29.22	1462	(-)-α-muurolene	null	4.61 ± 0.17 b	5.08 ± 0.12 a	4.78 ± 0.15 ab
30.01	1477	γ-Muurolene	woody, spice	1.16 ± 0.07 b	1.48 ± 0.06 a	1.21 ± 0.07 b
30.25	1481	β-Selinene	null	3.40 ± 0.08 b	3.98 ± 0.21 a	3.46 ± 0.15 b
30.49	1485	α-Muurolene	null	5.87 ± 0.23 a	5.89 ± 0.16 a	5.33 ± 0.27 b
31.02	1495	α-Farnesene	citrus, herbal, lavender	7.77 ± 0.13 b	8.05 ± 0.03 a	8.01 ± 0.36 a
31.52	1505	β-cadinene	green, woody	33.64 ± 1.17 a	31.85 ± 0.84 a	29.20 ± 1.07 b
31.86	1512	β-Bisabolene	balsamic, woody	2.23 ± 0.10 a	1.92 ± 0.15 b	2.30 ± 0.09 a
32.16	1519	Cubenene	null	7.12 ± 0.20 a	6.4 ± 0.24 b	5.81 ± 0.26 c
32.54	1527	α-Calacorene	woody	2.72 ± 0.19 a	1.63 ± 0.04 b	1.54 ± 0.12 b
34.37	1566	Caryophyllene oxide	sweet, fresh, woody	3.41 ± 0.26 a	2.58 ± 0.14 b	2.73 ± 0.23 b
35.41	1587	Trans-Z-α-Bisabolene epoxide	null	-	0.63 ± 0.08	-
		**Acid and Esters**				
9.56	1078	Heptanoic acid	waxy, cheesy, fruity	2.06 ± 0.10 a	1.35 ± 0.09 b	0.92 ± 0.07 c
14.23	1183	Octanoic acid	fatty, waxy, rancid	5.01 ± 0.23	-	3.72 ± 0.06
21.77	1319	Methyl geranate	waxy, green, fruity	8.97 ± 0.37 b	9.35 ± 0.74 a	9.32 ± 0.61 a
21.97	1324	Methyl decanoate	oily, fruity, floral	1.95 ± 0.16	-	-
23.29	1350	Citronellyl acetate	floral, waxy, aldehydic	5.47 ± 0.13 a	5.21 ± 0.43 a	4.42 ± 0.34 b
23.73	1358	Neryl acetate	floral, rosy, soapy	7.94 ± 0.40 b	9.58 ± 0.50 a	8.17 ± 0.64 b
23.98	1363	2-(2-Butoxyethoxy)ethyl acetate	null	6.16 ± 0.19 a	5.69 ± 0.28 b	5.23 ± 0.29 b
24.23	1367	Decanoic acid	soapy, waxy, fruity	1.00 ± 0.06 c	1.55 ± 0.08 b	2.25 ± 0.04 a
24.75	1377	Geranyl acetate	waxy, green, floral	14.66 ± 0.20 b	18.68 ± 0.82 a	14.59 ± 0.57 b
25.8	1395	Methyl methanthranilate	fruity, woody, floral	10.25 ± 0.36 b	11.03 ± 0.10 a	9.40 ± 0.46 c
31.68	1508	Dihydroactinidiolide	musk, coumarin	14.86 ± 1.90	16.94 ± 1.39	15.29 ± 0.98
34.94	1578	Pentanoicacid,2,2,4-trimethyl-3-Carboxyisopropyl,isobutylester	null	4.04 ± 0.26 a	1.85 ± 0.15 b	2.07 ± 0.16 b
42.43	1922	Methyl palmitate	oily, waxy, fatty, orris	6.89 ± 0.37 a	1.52 ± 0.18 c	4.15 ± 0.24 b
44.35	2089	Methyl linolenate	null	0.38 ± 0.05	-	0.21 ± 0.01
		**Others**				
7.86	1026	o-Cymene	null	6.89 ± 0.46 a	3.06 ± 0.05 b	6.50 ± 0.69 a
8.5	1047	1-ethylpyrrole-2-carbaldehyde	burnt, roasted, smoky	-	-	4.25 ± 0.26
9.98	1090	α,P-Dimethylstyrene	spicy, balsamic, musty	5.24 ± 0.38 a	3.91 ± 0.06 b	3.48 ± 0.16 c
12.89	1157	(+)-β-Pinene oxide	rosemary, sage, herbal	11.67 ± 0.24 a	6.55 ± 0.99 c	9.21 ± 0.41 b
19.59	1281	3,4-Diethylphenol	null	-	2.44 ± 0.08 a	1.43 ± 0.08 b
21	1303	1,2,3-Trimethoxybenzene	null	10.39 ± 0.86	9.51 ± 0.36	9.09 ± 0.42
24.16	1366	1,2,4-Trimethoxybenzene	null	13.90 ± 1.16	13.55 ± 1.02	12.6 ± 0.77
40.17	1768	3-methylheptadecane	null	0.56 ± 0.05 a	0.36 ± 0.01 b	0.37 ± 0.02 b

**Note:** Different lowercase letters in each row indicate significant difference at *p* < 0.05. HA, hot air fixing; ST, steam fixing; SL, sunlight fixing; RT, retention time; RI, retention index. Odor description found in the TGSC website (https://www.thegoodscentscompany.com/ (accessed on 10 October 2021)).

**Table 3 molecules-28-01079-t003:** Analysis of aroma species and contents in orange dark tea with different fixing methods (μg/g).

Samples	HA	ST	SL
Alcohols	647.48 ± 24.08 a	459.93 ± 44.57 c	555.94 ± 36.30 b
Aldehydes and Ketones	472.68 ± 24.95 a	405.97 ± 19.99 b	385.31 ± 23.68 b
Alkenes	403.84 ± 27.23 b	467.34 ± 22.51 a	387.24 ± 9.17 b
Acid and Asters	89.65 ± 4.77 a	82.74 ± 4.74 ab	79.73 ± 4.48 b
Others	48.64 ± 3.15 a	39.38 ± 2.57 b	46.92 ± 2.80 a
Total	1662.29 ± 84.20 a	1455.36 ± 94.40 b	1455.14 ± 76.45 b

**Note:** Different lowercase letters in each row indicate significant difference at *p* < 0.05. HA, hot air fixing; ST, steam fixing; SL, sunlight fixing.

**Table 4 molecules-28-01079-t004:** Antioxidant activity of ODT samples with different fixing method.

Samples	μmol TE/g			APC Index			Comprehensive APC Index
	FRAP	DPPH	ABTS	FRAP	DPPH	ABTS	
HA	254.49 ± 1.84 a	351.11 ± 3.21 a	324.04 ± 1.07 ab	100.00	100.00	99.39	99.8
ST	224.91 ± 1.12 b	334.33 ± 1.94 b	322.29 ± 1.23 b	88.37	95.22	98.85	94.15
SL	212.61 ± 8.67 c	344.41 ± 5.07 a	326.04 ± 2.32 a	83.54	98.09	100.00	93.88

**Note:** Different lowercase letters in each column indicate significant difference at *p* < 0.05. HA, hot air fixing; ST, steam fixing; SL, sunlight fixing. FRAP, ferric ion reducing antioxidant capacity; DPPH, free radical scavenging property by diphenyl-1-picrylhydrazyl radical; ABTS, free radical scavenging property by 2,2-azino-bis-3-ethylbenzothiazoline-6-sulfonic acid radical.

**Table 5 molecules-28-01079-t005:** Inhibitory effect of different fixing methods on ODT enzyme activity (IC_50_ μg/mL).

	α-Glucosidase	α-Amylase
HA	387.59 ± 7.50 c	780.43 ± 24.99 a
ST	401.67 ± 4.22 b	538.17 ± 16.25 b
SL	438.11 ± 6.99 a	514.16 ± 13.42 b

**Note:** Different lowercase letters in each column indicate significant difference at *p* < 0.05. HA, hot air fixing; ST, steam fixing; SL, sunlight fixing.

**Table 6 molecules-28-01079-t006:** Correlation analysis between ODT functional activities and chemical components.

Substances	FRAP	DPPH	ABTS	α-Amylase	α-Glucosidase
Amino acid	0.202	0.235	0.162	0.201	−0.139
Soluble sugar	0.025	−0.574	−0.679 *	−0.218	−0.435
Tea polyphenols	0.848 **	0.698 *	−0.028	0.886 **	−0.642
Theaflavins	−0.792 *	−0.752 *	−0.061	−0.943 **	0.507
Thearubin	−0.785 *	−0.802 **	−0.197	−0.957 **	0.470
Theabrownin	−0.350	−0.273	0.022	−0.246	0.240
Soluble protein	−0.818 **	−0.129	0.576	−0.557	0.943 **
Flavonoids	−0.346	0.492	0.754 *	0.069	0.720 *
Polyphenols	−0.152	0.618	0.792 *	0.233	0.589
Polysaccharide	−0.683 *	−0.926 **	−0.314	−0.932 **	0.278
Hesperidin	−0.071	0.657	0.691 *	0.336	0.492
Synephrine	0.199	0.779 *	0.743 *	0.528	0.250
Limonin	−0.600	0.228	0.763 *	−0.249	0.886 **

**Note:** * and ** indicate significant correlation at *p* < 0.05 and extremely significant correlation at *p* < 0.01, respectively.

**Table 7 molecules-28-01079-t007:** Sensory evaluation of ODT samples at different fixing temperatures.

Sample	Appearance (10 Points)	Soup Color (10 Points)	Aroma (30 Points)		Taste (50 Points)					Total Score
			Aroma	Coordination	Sour	Sweet	Bitter	Astringency	Coordination	
70 °C	7.60 ± 0.400 a	9.10 ± 0.10	15. 00 ± 0.00	9.00 ± 0.00	7.00 ± 0.00 a	7.00 ± 0.00	5.50 ± 0.30 c	5.50 ± 0.30 cd	5.90 ± 0.30 c	71.60 ± 0.90 c
75 °C	6.50 ± 0.00 b	9.00 ± 0.10	14.90 ± 0.00	9.00 ± 0.00	6.70 ± 0.60 ab	7.00 ± 0.00	5.70 ± 0.00 c	5.80 ± 0. 00 d	6.30 ± 0.10 d	70.90 ± 0.70 c
80 °C	6.30 ± 0.30 bc	9.10 ± 0.10	14.90 ± 0.00	9.00 ± 0.00	7.00 ± 0.00 a	7.00 ± 0.00	6.70 ± 0.60 ab	6.70 ± 0.60 ab	6.80 ± 0.30 bc	73.50 ± 1.80 ab
85 °C	6.40 ± 0.20 bc	9.10 ± 0.10	15.10 ± 0.10	9.00 ± 0.00	6.50 ± 0.00 c	7.00 ± 0.00	6.80 ± 0.30 ab	6.90 ± 0.10 a	6.90 ± 0.10 b	73.70 ± 0.60 ab
90 °C	6.5 ± 0.3 b	9.1 ± 0.1	15 ± 0.0	9.0 ± 0.0	6.8 ± 0.0 b	7.0 ± 0.0	7.0 ± 0.0 a	7.0 ± 0.0 a	7.0 ± 0.0 b	74.5 ± 0.2 a
95 °C	6.3 ± 0.3 bc	9.1 ± 0.1	15 ± 0.0	9.0 ± 0.0	6.8 ± 0.0 b	7.0 ± 0.0	7.0 ± 0.0 a	7.0 ± 0.0 a	7.5 ± 0.0 a	74.7 ± 0.4 a
100 °C	6.1 ± 0.1 c	9.1 ± 0.2	14.9 ± 0.1	9.0 ± 0.0	6.8 ± 0.0 b	7.0 ± 0.0	7.0 ± 0.0 a	7.0 ± 0.0 a	7.5 ± 0.0 a	74.4 ± 0.2 a

**Note:** Different lowercase letters in each column indicate significant difference at *p* < 0.05.

## Data Availability

Not applicable.

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
