# Peer review of "The New Insight into the Effects of Different Fixing Technology on Flavor and Bioactivities of Orange Dark Tea"

_molecules, 2023, doi:10.3390/molecules28031079_

Round 1

Reviewer 1 Report

In this manuscript, the authors evaluate the effect of different fixation techniques on the flavor and bio-activity of orange dark tea. This work has clear ideas, detailed data, and has a certain guiding significance for tea processing. Therefore, I suggest that it be accepted after minor revision.

Improvements: Please keep the same number of decimal places in all Tables.

Author Response

To Reviewer #1:

Reviewer: 1

Comments to the Author

In this manuscript, the authors evaluate the effect of different fixation techniques on the flavor and bio-activity of orange dark tea. This work has clear ideas, detailed data, and has a certain guiding significance for tea processing. Therefore, I suggest that it be accepted after minor revision.

Improvements: Please keep the same number of decimal places in all Tables.

Reply: Thank you for the suggestion. We have checked the whole manuscript and corrected the decimal points for consistency.

Reviewer 2 Report

TITLE: The new insight into the effects of different fixing technology on flavor and bioactivities of orange dark tea

 Comments to authors:

This study compared the quality of orange dark tea fixed by hot air, steam and traditional sunlight, and the hot air-fixed orange dark tea was shown to be in good harmony with the aroma of the fruit and the tea, with a mellow taste and high sensory quality, and this study reveals for the first time the effect of the fixing method on the antioxidant activity of orange dark tea and the inhibition of α-glucosidase and α-amylase activities. However, the manuscript needs revisions and the suggestions are revealed as follows.

1.       The innovation of this paper is weak, and the experimental material selected is very niche, which is not of great guiding significance to the actual production.

2.       The introduction fails to reflect the key and innovative points of the paper, so it is suggested to introduce the unique biological activity and flavor of peach leaf orange dark tea, and further explain the reasons for choosing peach leaf orange dark tea.

3.       Please provide relevant qualifications of sensory reviewers to ensure the accuracy of experimental results.

4.       The moisture of the orange dark tea with different fixing methods should be provided.

5.       It is suggested to add odor descriptions of different volatile components in Table 2.

6.       It is suggested to add electronic tongue to detect the effect of fixing method on taste.

7.       Please unify the width and format of the column chart in Figure 2.

8.       The color of lines representing different temperatures in Figure 4 cannot be clearly distinguished, so it is recommended to modify.

9.       Please mark the significant difference of the data in Figure 5.

10.   There are some grammars errors in the manuscript, please check throughout the manuscript and to ensure the grammars are correct.

 Author Response

To Reviewer #2:

Reviewer: 2                                                                                                   

Comments to the Author

This study compared the quality of orange dark tea fixed by hot air, steam and traditional sunlight, and the hot air-fixed orange dark tea was shown to be in good harmony with the aroma of the fruit and the tea, with a mellow taste and high sensory quality, and this study reveals for the first time the effect of the fixing method on the antioxidant activity of orange dark tea and the inhibition of α-glucosidase and α-amylase activities. However, the manuscript needs revisions and the suggestions are revealed as follows.

  1. The innovation of this paper is weak, and the experimental material selected is very niche, which is not of great guiding significance to the actual production.

Reply: We thank the reviewer for the comment. This experiment involved two materials, peach leaf orange and dark tea. Peach leaf orange is a characteristic citrus germplasm resource on both sides of the Three Gorges of the Yangtze River in Hubei Province, with a planting area of more than 3000 ha, coupled with more than 10,000 growers, and is the main economic source for these rural populations. The fruit used in the present study is the secondary fruit, with a very low price for fresh fruit, but when the peel was processed into orange dark tea, the fruit price can be increased by more than 50%, thus significantly increasing the income of local farmers. The dark tea used for the experiment was Hubei Qingzhuan tea, which is a representative dark tea in China with the production area covering several tea areas in Xianning, Enshi, Yichang and Huanggang, invovling a tea plantation area of over 70,000 ha and a population of over 200,000 people. Processing peach leaf orange and Qingzhuan tea into peach leaf orange dark tea can increase the price by more than 50% and significantly increase the income of farmers and enterprises, suggesting that this study has great guiding significance for practical production.

  1. The introduction fails to reflect the key and innovative points of the paper, so it is suggested to introduce the unique biological activity and flavor of peach leaf orange dark tea, and further explain the reasons for choosing peach leaf orange dark tea.

Reply: We thank the reviewer for the comment. We have added the functions of peach leaf orange peel and Qingzhuan tea and the flavor characteristics of peach leaf orange dark tea in the introduction part. The development of peach leaf orange dark tea was first carried out by the authors and the product is very popular among consumers. Their combination complemented each other in function, also suggesting the novelty of this study.

  1. Please provide relevant qualifications of sensory reviewers to ensure the accuracy of experimental results.

Reply: We thank the reviewer for the comment. The personnel involved in the sensory evaluation were qualified tea evaluators with certificate numbers 1547073077300236 (Jingtao Zhou), 2017000000112534 (Junyu Zhu), and 1617021905301656 (Ming Dai), and scanned copies of the certificates were uploaded as attachment for reference. In addition, the evaluation was also conducted under the guidance of relevant professors in this field, and the results were reliable.

  1. The moisture of the orange dark tea with different fixing methods should be provided.

Reply: Thanks for the suggestion. We have submitted the data on the moisture content of orange tea with different fixing treatments as Supplementary Material Table S1, corresponding to lines 113-114 in the manuscript.

  1. It is suggested to add odor descriptions of different volatile components in Table 2.

Reply: Thanks for the suggestion. We have added the aroma description of volatile substances in Table 2.

  1. It is suggested to add electronic tongue to detect the effect of fixing method on taste.

Reply: Thanks for the suggestions. Regarding the judgment of tea quality, sensory review is the main focus both at home and abroad, and national standards have been established. There are few reports in the literature that use only electronic nose to judge taste quality, and it is only used as an auxiliary tool. In addition, the taste review in this experiment also included coordination, and based on the results we obtained in the school and the testing institute in Wuhan city, the electronic nose in this area could not judge coordination. Therefore, the quality judgment in this experiment is mainly based on sensory evaluation. In the subsequent study, we will add electronic nose if any better performance could be achieved.

Chinese national standards < Methods of Sensory Evaluation of Tea > (GB/T 23776-2018)

  1. Li, Q.; Jin, Y.; Jiang, R.; Xu, Y.; Zhang, Y.; Luo, Y.; Huang, J.; Wang, K.; Liu, Z., Dynamic changes in the metabolite profile and taste characteristics of Fu brick tea during the manufacturing process. Food Chemistry 2021, 344, 128576.
  2. Chen, L.; Liu, F.; Yang, Y.; Tu, Z.; Lin, J.; Ye, Y.; Xu, P., Oxygen-enriched fermentation improves the taste of black tea by reducing the bitter and astringent metabolites. Food Research International 2021, 148, 110613.
  3. Liu, H.; Zhuang, S.; Gu, Y.; Shen, Y.; Zhang, W.; Ma, L.; Xiao, G.; Wang, Q.; Zhong, Y., Effect of storage time on the volatile compounds and taste quality of Meixian green tea. LWT 2023, 173, 114320.
  4. Please unify the width and format of the column chart in Figure 2.

Reply: Thanks for the suggestion. We have unified the column width of Figure 2.

  1. The color of lines representing different temperatures in Figure 4 cannot be clearly distinguished, so it is recommended to modify.

Reply: Thanks for the comment. We have modified Figure 4 as suggested.

  1. Please mark the significant difference of the data in Figure 5.

Reply: Thanks for the suggestion. We have marked the significant differences in Figure 5 as suggested.

  1. There are some grammars errors in the manuscript, please check throughout the manuscript and to ensure the grammars are correct.

Reply: Thanks for the suggestion. We have checked the manuscript carefully in grammar and spelling. And the manuscript was also modified by a professional English teacher and a native speaker.

Author Response

Dear editor and reviewers:

We sincerely thank you and all the reviewers for the valuable comments and suggestions on how to improve the quality of our manuscript. We have carefully read the comments and revised the manuscript as suggested. The responses to all the comments are listed below one by one in blue text, and changes in the revised manuscript are given in red text.

 To Reviewer #3:

Reviewer: 3

Comments to the Author

Dear Editor,

The article “The new insight into the effects of different fixing technology on flavor and bioactivities of orange dark tea (2160126)” that I reviewed. This article is very interesting. It compares the effects of different processes on the flavor and functional activity of orange peel black tea, which is of significance to the optimization of production process. However, there are some unclear points that need further explanation before published.

  1. The text framework of temperature captions in Figure 4 is incomplete. Please check and modify it.

Reply: Thanks for the comment. We have replaced Figure 4 with a more complete graph.

  1. In the line 153, the subscript and superscript of R2Y, Q2 and IC50 is not standard.

Reply: Thank you for pointing out the formatting issues. The written format has been revised in the article.

  1. Similarly, in the section 2.4, the format of the sentence “Effects of different fixing methods on bioactivities:IC50” is standard. Please check these similar problems in the whole manuscript.

Reply: Thank you for pointing out the formatting issues. The written format has been revised in the article.

  1. Statistical analysis: the analysis method is not complete. For example, the drawing software information is missing.

Reply: We thank the reviewer for the suggestions. Information related to data analysis and mapping has been added to the article.

  1. How do you calculate antioxidant potency composite (APC) value? The authors just cited a reference about it, but the calculation details are missing. Please specify the calculation details of APC.

Reply: Thanks to the reviewer for the suggestion. The references for the APC calculation are as follows, APC is calculated by the equation:

APC = (Index DPPH 1+ Index ABTS+Index FRAP) / 3

Reference: Seeram, N.P.; Aviram, M.; Zhang, Y.; Henning, S.M.; Feng, L.; Dreher, M.; Heber, D. Comparison of antioxidant potency of commonly consumed Polyphenol-Rich beverages in the united states. Journal of Agricultural and Food Chemistry 2008, 56, 1415-1422, doi:10.1021/jf073035s.

  1. Different from the traditional sun-drying, why only two fixation ways of hot air and steaming blanching are selected in this research?

Reply: Thanks to the reviewer for the question. These two processes have been used in citrus tea production, and no process optimization has been performed yet, and it is not clear which fixing method is more suitable for citrus tea processing. This study is aimed to optimize the production process to determine a better fixing process.

  1. Materials and Methods: What is the basis for selecting the process parameters of fixation and drying? Why do you choose such parameters? The fixation temperature of the green tea usually exceeds 200 ℃, and the temperature of the three fixation methods in the article did not exceed 100 ℃. Please explain the reason you adopted such a low fixation temperature.

Reply: We thank the reviewer for the questions. We surveyed many citrus tea processing plants before the experiment to collect the process parameters of these plants which can be adopted in this experiment. The "fixing" in this study is completely different from the green tea fixing process. The tea leaves in orange dark tea are dried tea leaves, which cannot be fixed at an extremely high temperature, otherwise coking will occur; an extremely high temperature will also deform the citrus shell, which does not meet the appearance requirements of citrus tea; moreover, high temperature may also affect the biological activity of citrus tea.

  1. For the common flavonoids in orange peel, only hesperidin was detected. Is there any limitation in the results?

Reply: Thanks for the question. The experimental design also included the determination of neohesperidin and naringin, but unlike other oranges, neohesperidin and naringin in peach leaf orange peel is very low and undetectable (see the figure below), so only the data for hesperidin were maintained.

Please do not hesitate to contact us if you have any questions about the manuscript. Thank you very much.

Kind regards
